# Whole cervix imaging of collagen, muscle, and cellularity in term and preterm pregnancy

Wenjie Wu [1,2], Zhexian Sun [1,2], Hansong Gao [2,3], Yuan Nan[2,3], Stephanie Pizzella[2], Haonan Xu[2,4], Josephine Lau[1,2], Yiqi Lin [2,3], Hui Wang [5], Pamela K. Woodard[1,6], Hannah R. Krigman[7], Qing Wang [1,4,6] ✉ & Yong Wang [2,6] ✉

Cervical softening and dilation are critical for the successful term delivery of a fetus, with premature changes associated with preterm birth. Traditional clinical measures like transvaginal ultrasound and Bishop scores fall short in predicting preterm births and elucidating the cervix's complex microstructural changes. Here, we introduce a magnetic resonance diffusion basis spectrum imaging (DBSI) technique for non-invasive, comprehensive imaging of cervical cellularity, collagen, and muscle fibers. This method is validated through ex vivo DBSI and histological analyses of specimens from total hysterectomies. Subsequently, retrospective in vivo DBSI analysis at 32 weeks of gestation in ten term deliveries and seven preterm deliveries with inflammation-related conditions shows distinct microstructural differences between the groups, alongside significant correlations with delivery timing. These results highlight DBSI's potential to improve understanding of premature cervical remodeling and aid in the evaluation of therapeutic interventions for at-risk pregnancies. Future studies will further assess DBSI's clinical applicability.

During a typical pregnancy, cervical softening commences as early as the first trimester and progresses throughout the majority of the third trimester, all while maintaining tissue integrity[1–4]. This is followed by an accelerated ripening phase occurring a few weeks or days prior to labor. Finally, in conjunction with regular uterine contractions, the ripened cervix dilates, facilitating term delivery of the fetus[1].

The cervix is composed of an extracellular matrix (ECM) containing fibrillar collagen (types I and III), elastin, and proteoglycans, alongside cellular components including fibroblasts, smooth muscle cells, and immune cells[5,6]. This composition is critical for the cervix's structural integrity. Progressive remodeling of the cervical ECM leads to restructured collagen fibrils and a consequent reduction in tensile strength. Predominantly, quantitative assessments of cervical remodeling have focused on variations in collagen, smooth muscle, and cellularity. Cervical softening is largely caused by increased hydration, disorganization of collagen fibers, increased collagen solubility, and decreased collagen concentration[7]. For example, analysis of human biopsy specimens from various gestational ages[8], immediately post-hysterectomy, and postpartum[9,10] revealed that, in the third trimester, more than 80% of the cervical collagen becomes soluble, and the collagen concentration is nearly half of that in the nonpregnant state. Smooth muscle cells in the cervix may also play a role in cervical

[1]Department of Biomedical Engineering, Washington University, St. Louis, MO, USA. [2]Department of Obstetrics & Gynecology, Washington University School of Medicine, St. Louis, MO, USA. [3]Department of Electrical & Systems Engineering, Washington University, St. Louis, MO, USA. [4]Department of Mechanical Engineering and Materials Science, Washington University, St. Louis, MO, USA. [5]Department of Physics, Washington University, St. Louis, MO, USA. [6]Mallinckrodt Institute of Radiology, Washington University School of Medicine, St. Louis, MO, USA. [7]Department of Pathology & Immunology, Washington University School of Medicine, St. Louis, MO, USA. ✉e-mail: wangqing@wustl.edu; wangyong@wustl.edu

remodeling and may act as a sphincter to regulate cervical closing and opening[11–13]. Both longitudinal and circumferential muscle fibers have been detected in the human cervix in histological imaging[12]. One hypothesis is that contraction of longitudinal muscle fibers promotes cervical dilation, whereas contraction of circumferential muscle fibers keeps the cervix closed[14]. Finally, immune cell infiltration could play a role in normal cervical remodeling. Early investigations of human biopsy specimens and rat models suggested that infiltration of inflammatory cells contributes to disintegration and disorganization of the collagen matrix[15–17]. Conversely, more recent studies in mice suggested that immune cell infiltration are not necessary to initiate cervical ripening, but instead participate in postpartum tissue repair[18,19]. Nonetheless, studies in mouse models suggested that infection and inflammation can promote premature cervical ripening[20,21], and Timmons et al. concluded that inflammatory cells are capable of inducing cervical ripening[1].

Several methods have been developed to quantitatively assess cervical remodeling at the organ level. Cervical elastography, which can be included as an adjunct to routine ultrasound imaging, was used to measure the cervical elasticity index[22,23]. However, accuracy of this method is affected by a patient's respiration, arterial pulsation, fetal movements, and operator hand motion. Other approaches have included acoustic attenuation measurement in pregnant humans[24], quantitative ultrasound measurements on nonpregnant human ex vivo specimens[25], and magnetic resonance elastography to measure elasticity and viscosity of non-pregnant cervices in vivo[26]. Several other imaging techniques are also being explored, such as optical coherence tomography for quantifying collagen fiber dispersion in ex vivo cervix specimens[27,28], Raman spectroscopy for measuring spectral changes on the ectocervix surface in vivo during gestation and labor[29,30], shear wave imaging for measuring changes in shear wave speed reflecting in vivo human cervix softness[3,31,32], and second harmonic generation imaging in ex vivo mouse specimens[33]. Several studies also indicate spatial heterogeneity in cervical softness, underscoring the need for comprehensive three-dimensional (3D) imaging of the entire cervix[26,31,32,34].

Several investigators have employed MR imaging (MRI) to quantify the changes in cervical microstructure in vivo. For example, Masselli et al. used MR diffusion weighted imaging to measure cervical hydration[35], and others have used 3D tractography of DTI data to delineate longitudinal and circumferential fiber tracts in the human cervix and uterus[36,37]. Qi et al. used a form of DTI that includes a tensor to describe the strength and directionality of water molecule diffusion in each voxel. With this method, they described mean, axial, and radial diffusivity, and fractional anisotropy measures to reflect fiber organization and hydration[38]. However, as none of these MRI techniques are able to differentiate and quantitate cellularity, collagen fibers, and muscle fibers within each imaging voxel, our understanding of the in vivo microstructure of the entire human cervix remains incomplete.

In current clinical practice, cervical length is used as a predictor of preterm birth, with studies indicating that early cervical shortening heightens the risk of preterm labor. For example, ultrasound imaging of cervical length during the second trimester[39,40] revealed that about 50% of women with a very short cervix ($\leq 15$ mm) delivered at 32 weeks of gestation or earlier. However, in another study with a different cutoff for diagnosing a short cervix, more than 60% of women with a cervix $\leq 25$ mm at 18–22 weeks' gestation delivered at full term ($\geq 37$ weeks)[41]. Cervical softening can be assessed manually to derive a Bishop score, but this method is not superior to cervical length as a predictor of preterm birth[42].

Addressing this critical gap in our understanding of cervical microstructures and monitoring cervical changes over gestation in 3D, this study describes a whole-cervix diffusion basis spectrum imaging (DBSI) multi-tensor model. Utilizing ex vivo specimens, we establish its ability to accurately image and quantitate collagen fibers, muscle fibers, and cells throughout the entire human cervix. This technique can provide voxel-level resolution of the cervix microstructural components, wherein each voxel is characterized by anisotropic and isotropic tensors to represent water diffusion within and around tissue-specific microstructural components. A whole brain DBSI multi-tensor model was originally validated for imaging brain inflammation in multiple sclerosis[43,44] and has been widely employed for brain microstructural imaging[45–47]. DBSI can be performed without contrast agents in a clinical MRI scanner and is safe for use during pregnancy[48–50]. Moreover, our analysis of in vivo DBSI data acquired at 32 weeks of gestation suggests that this method can be used to discern differences between the cervices of patients who delivered at term and those who had inflammation and delivered preterm. Consequently, DBSI holds great promise to provide in vivo biomarkers that accurately reflect cervix microstructure and remodeling. Such biomarkers could be instrumental in predicting preterm birth and assessing the efficacy of therapeutic strategies to prevent preterm birth in humans.

## Results
### Development and validation of whole cervix DBSI microstructure parameters

To design a realistic multi-tensor model for imaging cervical microstructures, we used Monte-Carlo simulation to compute the Brownian motion trajectories (dashed arrows in Fig. 1b) of randomly distributed water molecules in different components of cervical microstructures at 37 °C under a diffusion gradient. A cervix contains cells, fibrillar collagen, proteoglycans, hyaluronan elastin, and water[7]. Additionally, some studies have identified smooth muscle cells and fibers in the human cervix[11,12]. Thus, we developed DBSI multi tensor models to reflect five types of water diffusion in cervical microstructures: intracellular water, hindered water, free water, collagen fibers, and muscle fibers.

The diffusion of intracellular water, hindered water, and free water is all isotropic. To model intracellular water diffusion in various cell types (including resident and immune cells) in human cervix, we designed spherical models with radii of $5 - 12\,\mu m$, with random "seeds" representing water molecules placed inside (Fig. 1b, c). The Monte-Carlo simulation results yielded restricted isotropic diffusion with apparent diffusion coefficients (which are equal to axial diffusivity [AD] and radial diffusivity [RD] for isotropic diffusion) between $0.089 \times 10^{-3}$ and $0.612 \times 10^{-3}\,mm^2/s$. We then defined hindered water as isotropic diffusion of water molecules whose trajectories (green dashed arrows, Fig. 1b, c) are hindered by dense, organized collagen fibers (purple rods) and cells. We defined free water as isotropic diffusion of water molecules whose trajectories (blue dashed arrows, Fig. 1b, c) that move freely around loose and disorganized collagen fibers (pink rods) and cells. To model free water, we derived diffusivity of $3 \times 10^{-3}\,mm^2/s$ from known experimental values[51,52]. Additionally, isotropic tensors with the diffusivity exceeding $10 \times 10^{-3}\,mm^2/s$ were utilized to account for the intravoxel incoherent motion effect (Fig. S13). A total of 150 isotropic tensors were then designed by discretizing the four ranges of isotropic diffusivity values.

The diffusion of water surrounding collagen fibers and within muscle fibers is anisotropic. For incoherent anisotropic water diffusion around collagen fibers, we simulated the random walk trajectories of water molecules traveling within a crossing bundle of tightly packed solid cylindrical rods at angles ranging between 0 and 20 degrees (purple rods, Fig. 1b, d) to reflect various fiber orientations and dispersion patterns indicative of both normal and pathological conditions in the cervix. The resulting diffusion simulation yielded AD values of $1.371 \pm 0.121 \times 10^{-3}\,mm^2/s$ and RD values of $0.357 \pm 0.021 \times 10^{-3}\,mm^2/s$. For coherent anisotropic water diffusion within muscle fibers, we simulated the random walk trajectories of water molecules traveling inside a bundle of parallel hollow cylindrical tubes (Fig. 1b, d). The resulting diffusion simulation yielded AD values of $1.622 \pm 0.109 \times 10^{-3}\,mm^2/s$ and RD values of $0.537 \pm 0.034 \times 10^{-3}\,mm^2/s$. Nine anisotropic

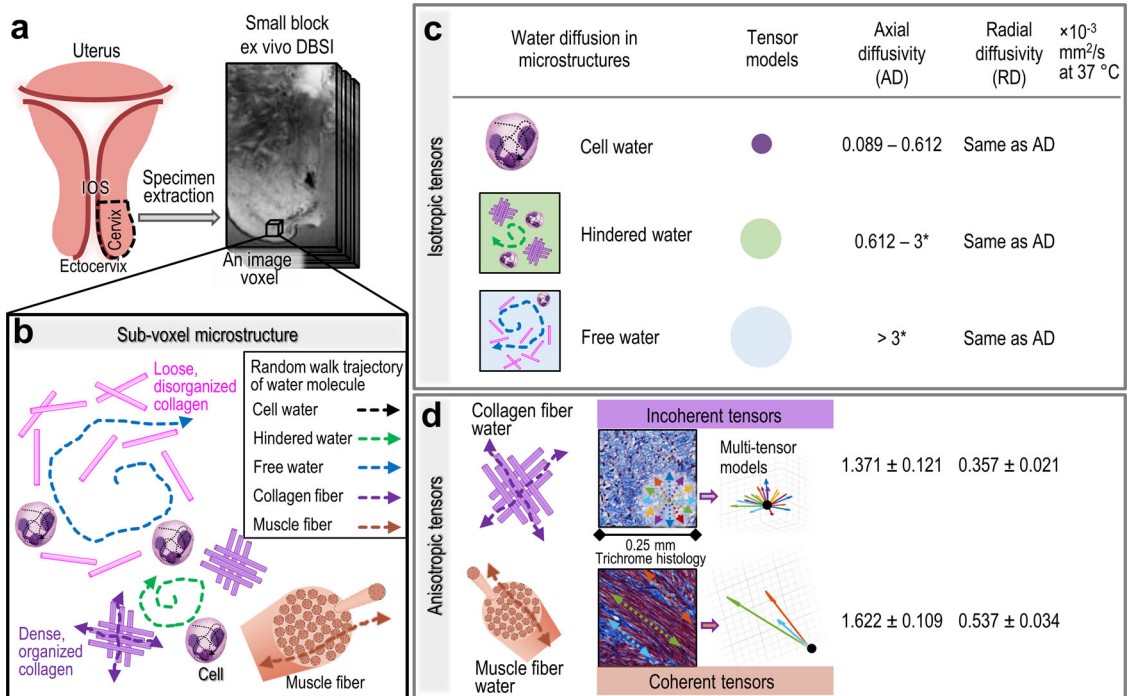

**Fig. 1 | The design of multi-tensor models for whole cervix DBSI based on the results from Monte-Carlo simulation. a** Schematic and representative magnetic resonance images of the cervical region used for ex vivo validation of the DBSI parameters. **b** In an image voxel in the cervix, we modeled three types of isotropic water diffusion: restricted isotropic diffusion within cell membranes (black dashed line in cells), hindered isotropic diffusion (green) around the dense, organized collagen fiber and cells, and free water diffusion (blue) in the region with loosely distributed and disorganized collagen fiber and cells. We also modeled two types of anisotropic water diffusion: incoherent anisotropic diffusion within bundles of tightly packed crossing collagen fibers (purple) and coherent anisotropic diffusion

inside bundles of parallel packed muscle fibers (coral). The dashed arrows are examples of water molecule trajectories under a diffusion gradient. **c** The isotropic tensor models are visualized as spherical balls with the radius reflecting their relative diffusivities. Axial and radial diffusivity of water molecules in cell, collagen, and muscle fiber from the Monte-Carlo simulation. *Free water diffusivity was derived from known experimental values[51,52]. **d** Anisotropic tensors represent water diffusion within a bundle of tightly packed collagen fibers (purple-colored crossing solid rods) and inside a bundle of parallel packed muscle fibers (coral-colored cylindrical tubes).

tensors (see Table S1 in the Supplementary Materials) were then designed to cover a broad range of these AD and RD values and were replicated in 25 principal directions uniformly distributed in a 3D space to achieve a total of 225 anisotropic tensors.

To validate the DBSI parameters, we recruited three patients who were undergoing planned total hysterectomies. Two multiparous patients (who provided samples P1-S1, P1-S2, and P2-S1) were both 37 years of age and were undergoing hysterectomy after delivery. Patient P1 had placenta previa and placenta accreta and underwent Caesarean hysterectomy at 34.2 weeks of gestation. The pathology examination showed endometriosis involving the cervix. Patient P2 had placenta accreta and underwent Caesarean hysterectomy at 34.0 weeks of gestation after spontaneous rupture of membrane. The pathology examination showed a normal cervix. One patient (sample NP1-S1) was 43 years of age, non-pregnant, and undergoing hysterectomy to treat long-term abnormal uterine bleeding. The pathology examination showed a cervix with chronic inflammation. After surgery, we obtained cervix specimens (Fig. 1a), embedded them in agar gel, and imaged with the DBSI sequence in a small animal MRI scanner. We then fixed the samples, cut sections, and stained them with hematoxylin and eosin (H&E) to detect nuclei and Masson's trichrome to detect collagen and muscle fibers. We then co-registered the stained sections and the MR images.

To compare cellularity in the histology and DBSI images, we segmented the H&E images and converted the positive stain of nuclei to the nuclei density maps (Fig. 2b). In the DBSI images, we calculated cell fraction as the percentage of diffusion signal contributed by intracellular water over total diffusion signals. Qualitatively, in all four specimens, the DBSI cell fraction maps and the histological maps showed that cells were most abundant in the subglandular region

(labeled SG in Fig. 2) near the endocervical canal. To quantitatively compare the DBSI-derived cell fraction and histological nuclei density maps, we downsampled the maps and performed correlation analysis by comparing the mean values in 2.5 × 2.5 mm grid boxes (solid white boxes, Fig. 2c). The DBSI-derived cell fractions correlated with histological nuclei densities in all four specimens (Fig. 2d).

In the trichrome-stained images (Fig. 3a), the positive stain of collagen (blue) was segmented and converted to collagen density maps (Fig. 3b). Qualitatively, the DBSI collagen fiber fraction maps and the histological collagen density maps showed the most abundant collagen in the subglandular region and the least abundant collagen in the outer stroma region, especially near the ectocervix. Quantitatively, the DBSI-derived collagen fiber fractions correlated with the histological collagen densities (Fig. 3d).

In the trichrome-stained images (Fig. 4a), the positive stain of muscle fibers (scarlet) was segmented and converted to muscle fiber density maps (Fig. 4b). In specimens P1-S1, P1-S2, and NP1-S1, the muscle was dense and clearly separated from the collagen. In the DBSI muscle fraction maps from these specimens, the orientations of muscle tracts appeared to coincide with those in the histological density maps. For example, specimen P1-S1 contained many longitudinal muscle fibers at middle locations radially from the canal, whereas specimen P1-S2 from the same patient contained many circumferential muscle fibers near the cervix-uterus junction (labeled "⊥"). Quantitatively, the DBSI-derived muscle fiber fractions correlated with the histological muscle densities (Fig. 4d). The strong correlations (Figs. 2d, 3d, and 4d) suggested that DBSI parameters allow us to accurately quantify cells, collagen, and muscle fibers throughout the entire human cervix samples. Additionally, our supplementary analysis

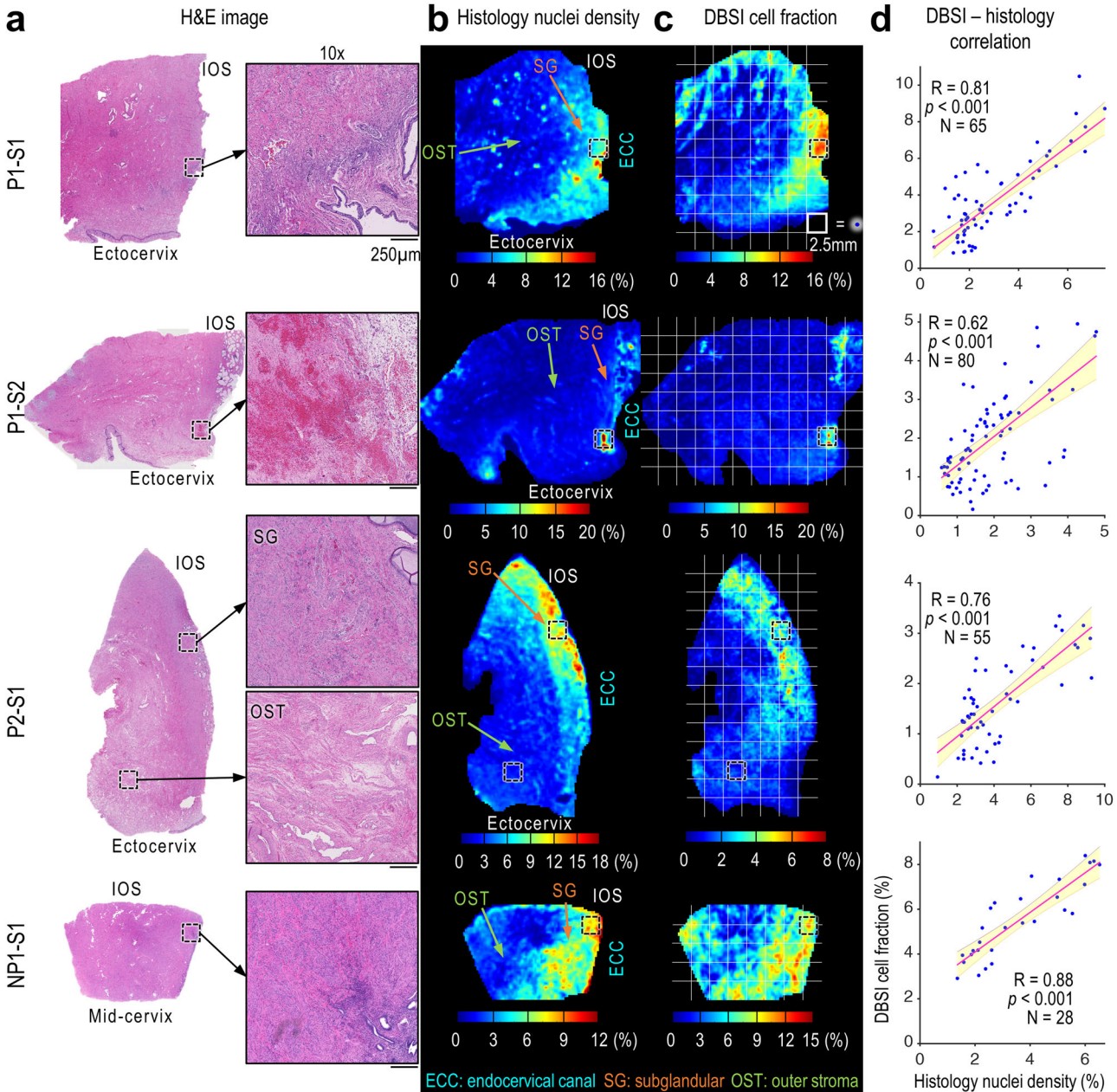

**Fig. 2 | Correlation between histological nuclei density maps and DBSI cell fraction maps. a** H&E images of cervix specimens and 10× magnified views of the black-dashed-boxed regions. The magnified views of P2-S1 show examples of high cell density in the subglandular (SG) region near the cervical canal and low cell density in outer stroma (OST) region, which are reflected on both histological density maps and DBSI maps. **b, c** Comparison of histological nuclei density maps (**b**) and DBSI-derived cell fraction maps (**c**). IOS, internal os; ECC, endocervical canal. **d** Pearson correlation coefficients (two-sided, at 0.05 significance level) of DBSI cell fraction and histology nuclei density. Each blue dot represents the mean value from a 2.5 × 2.5 mm white grid box within the specimen's contour in (**b**) and (**c**). The red lines are the linear fits, and the shaded areas are 95% confidence intervals. $p = 1.76 \times 10^{-16}$, $8.11 \times 10^{-10}$, $2.40 \times 10^{-11}$, $8.25 \times 10^{-10}$ from P1-S1 to NP1-S1, respectively. Source data are provided as a Source Data file.

with diffusion tensor imaging (DTI) and free water imaging (FWI) models (Figs. S5–S12) shows inconsistent correlations between histology-derived data, such as nuclei, collagen and muscle fiber density, and DTI and FWI metrics.

**In vivo DBSI measures of the cervix differ between term and preterm patients**

To begin to assess the utility of our whole cervix DBSI, we retrospectively analyzed DBSI data collected from patients at $32 \pm 2$ weeks of gestation who were enrolled in other studies. For this study, we identified ten healthy patients from the parent study who had

delivered at term (at or after 37 weeks of gestation) and did not present any adverse pregnancy complications. Eight of the ten patients were admitted for induced term labor. Two patients were admitted for spontaneous term labor with contractions, one of whom presented with rupture of membrane at admission. Additionally, we identified ten patients in total from the parent study who completed the 32-week MR imaging, had inflammation-related adverse conditions (described in the method section in detail) and delivered preterm (less than 37 weeks of gestation). Three of the ten preterm patients were excluded because of inadequate image quality and severe motion blur; thus, seven preterm patients with adverse inflammatory conditions were

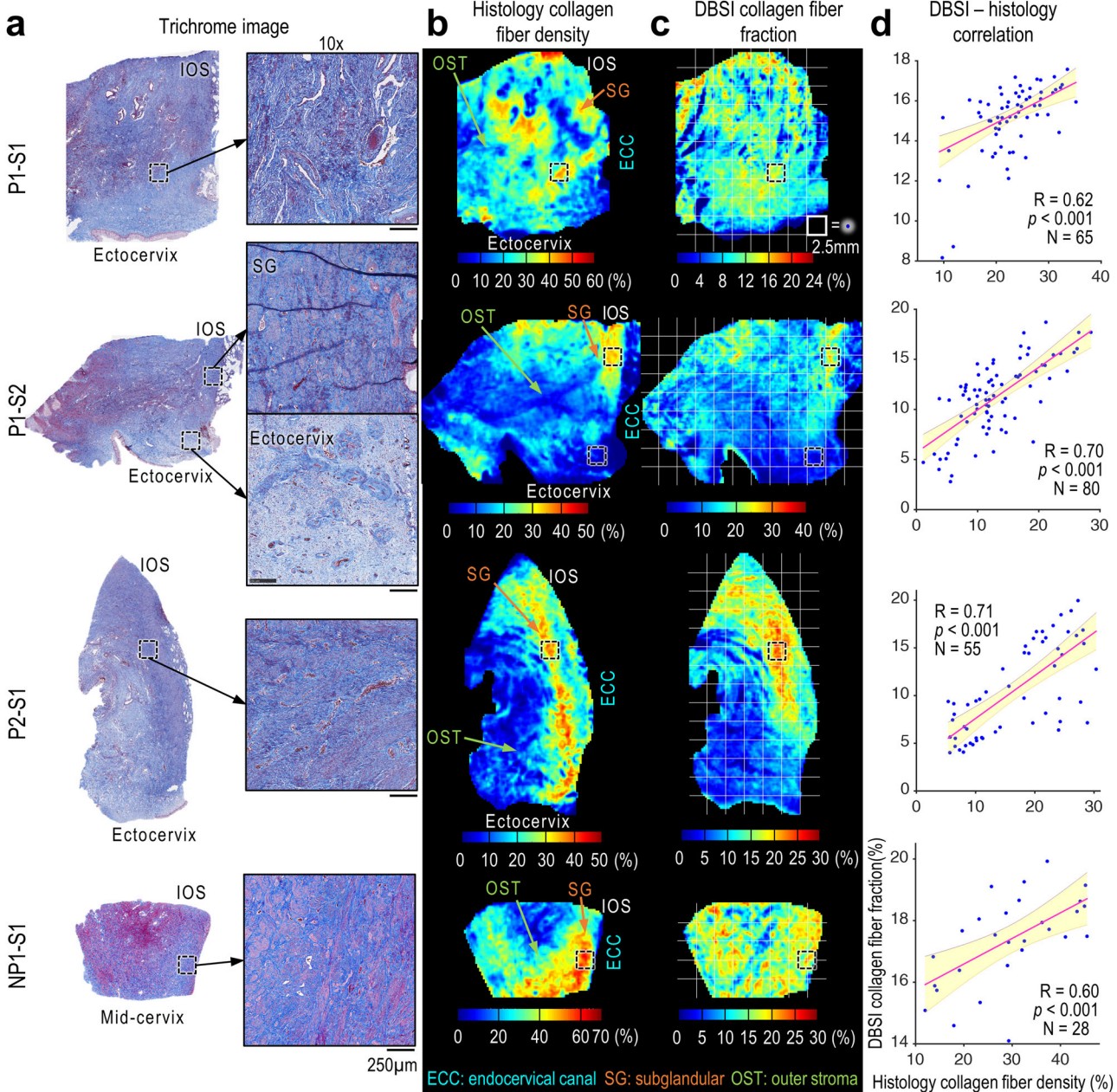

**Fig. 3 | Correlation between histological collagen density maps and DBSI collagen fraction maps. a** Trichrome images of the specimens and 10× magnified views of the black-dashed-boxed regions. The magnified views of P1-S2 show examples of dense organized collagen fibers in the subglandular (SG) region and loose disorganized collagen fiber near the ectocervix, which are reflected on both histological density and DBSI maps. **b, c** Comparison of histological collagen density maps (**b**) and DBSI-derived collagen fraction maps (**c**). **d** Pearson correlation coefficients (two-sided, at 0.05 significance level) between DBSI collagen fiber fraction and histology collagen fiber density. Each blue dot represents the mean value from a 2.5 × 2.5 mm white grid box within the specimen's contour in (**b**) and (**c**). Red lines indicate linear fits, and shaded areas indicate 95% confidence intervals. $p = 4.51 \times 10^{-8}$, $3.30 \times 10^{-13}$, $1.65 \times 10^{-9}$, $6.57 \times 10^{-4}$ from P1-S1 to NP1-S1, respectively. Source data are provided as a Source Data file.

included in the analysis (Table 1). In the preterm group, two patients were admitted for spontaneous preterm labor with contraction, one of whom presented with rupture of membrane at admission; three patients were admitted for induced labor due to preeclampsia and delivered vaginally preterm; one patient was admitted for preterm Caesarean section due to preeclampsia; and one patient was diagnosed with endometriosis in the cervix and was admitted for preterm Caesarean section due to placenta previa and accreta. The term and preterm groups were similar with regard to maternal age and race (Table 1).

Our in vivo DBSI data distinguishes between term and inflammation-associated preterm deliveries. This is accomplished by quantifying DBSI metrics like collagen fraction, which involves calculating the fraction of the T2-weighted signal from diffused water molecules among collagen fibers, relative to the voxel's total T2-weighted signal under diffusion gradients. This approach is similarly applied to DBSI-cell and -muscle fractions. Figure 5 shows T2-weighted (T2W) MR images and DBSI cellularity, collagen fiber, and muscle fiber fraction maps of the cervix in four representative preterm patients and four representative term patients. We used the T2W images to

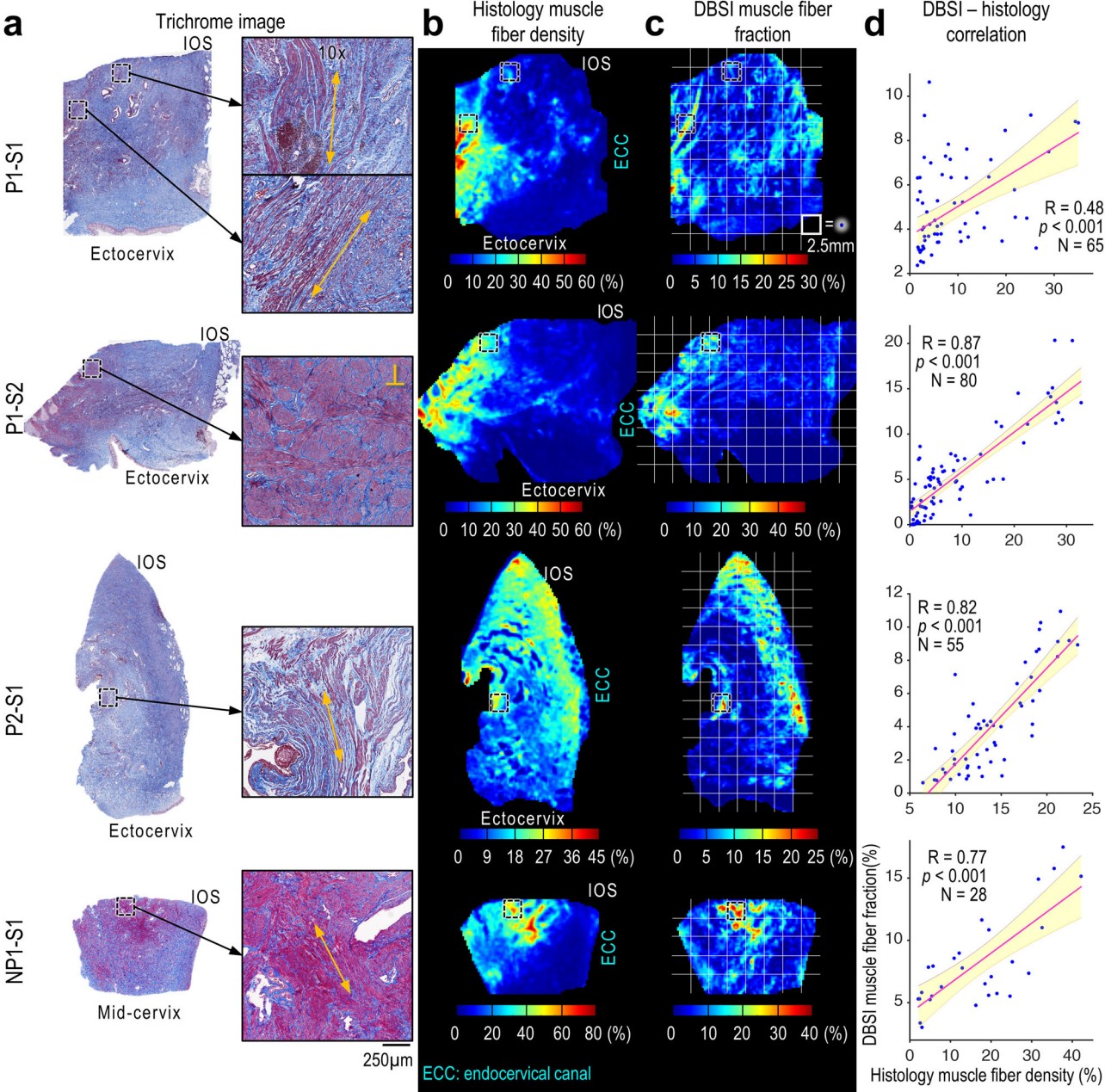

**Fig. 4 | Correlation between histological muscle density maps and DBSI muscle fraction maps. a** Trichrome images of the specimens and 10× magnified views of the black-dashed-boxed regions. Specimen P1-S1 shows many longitudinal muscle fibers at middle locations radially from the canal; fiber directions indicated by yellow arrows). Specimen P1-S2 shows many circumferential muscle fibers near the cervix-uterus junction that are perpendicular (yellow⊥) to the slides. **b**, **c** Comparison of histological muscle density maps (**b**) and DBSI-derived muscle fraction maps (**c**). **d** Pearson correlation coefficients (two-sided, at 0.05 significance level) of DBSI muscle fiber fraction and histology muscle fiber density. Each blue dot represents the mean value from a 2.5 × 2.5 mm white grid box within the specimen's contour in (**b**) and (**c**). Red lines are linear fits and shaded areas are 95% confidence intervals. $p = 5.88 \times 10^{-5}$, $2.07 \times 10^{-25}$, $2.98 \times 10^{-14}$, $1.85 \times 10^{-6}$ from P1-S1 to NP1-S1, respectively. Source data are provided as a Source Data file.

segment the cervix images and mask the mucus-filled endocervical canals. Qualitatively, the cervices from all four preterm patients exhibited higher DBSI cell fraction (with a few hot spots at 10%) than did the cervices from the term patients, in which the DBSI cell fractions were less than 5%. Cervices from the preterm patients appeared to have lower collagen density (less than 5% in most areas) than the cervices from the term patients (10%–15% in most areas). The cervices from the preterm patients had several regions in which the muscle fractions were nearly 20%, whereas the cervices from the term patients had somewhat lower muscle fraction. Finally, the cervices in the preterm patients appeared to have higher DBSI free water fraction (up to 60%) than did the cervices in the term patients (mostly below 30%).

Next, we quantitatively compared the DBSI measures in the whole cervical volume from all 17 patients. Compared to the cervices from the term group, the cervices from the preterm group had significantly higher median DBSI cell fraction (Fig. 6a), significantly lower median collagen fiber fraction (Fig. 6b), significantly higher median muscle fiber fraction (Fig. 6c), and significantly higher median free water fraction (Fig. 6d) in the entire cervical volume. Specifically, in the preterm group, the median values for cell fraction, muscle fraction,

and free water fraction are approximately 1.94, 1.46, and 1.36 times of those in the term group, respectively. Conversely, the median collagen fraction in the term group is about 1.65 times that of the preterm group. We plotted the median DBSI measures of cell fraction, collagen fiber fraction, and muscle fiber fraction on X-Y-Z axes for each patient and generated 3D Gaussian ellipsoids fit with two standard deviations of the mean and 95% probability in each dimension. We observed a clear separation between the term and preterm ellipsoids (Fig. 6o).

Finally, we examined data with regard to days from DBSI imaging until delivery. We found significant linear correlations between days to delivery and median DBSI-cell (Fig. 6e), -collagen fiber (Fig. 6f), and -muscle fiber (Fig. 6g) fractions. We also found significant linear correlations between these DBSI measures and cervical lengths measured at 32 ± 2 weeks of gestation in T2W MR images (Fig. 6i–k). In contrast, no correlation was found between DBSI free water fraction and either days to delivery (Fig. 6h) or cervical length at 32 weeks (Fig. 6l). The cervical lengths, measured at 32 ± 2 weeks of gestation in the T2W MR images (Fig. 6m), significantly differed between the term and preterm groups as expected. In contrast, the cervical lengths measured at 20 ± 2 weeks of gestation by transvaginal ultrasound (Fig. 6n) did not significantly differ between the term and preterm groups. Taken together, these data suggest that the DBSI-derived measures of the cervix at 32 ± 2 weeks' gestation can differentiate between patients who deliver at term and patients who deliver preterm.

## Discussion

In this study, we developed and validated whole cervix DBSI measures to visualize and quantify microstructural features in vivo during human pregnancy. Our ex vivo results demonstrated that DBSI-derived cellularity, collagen fiber, and muscle fiber fraction maps correlated spatially with their corresponding histological maps. Furthermore, our in vivo data suggests that our DBSI-derived measures from the whole cervical volume, collected around 32 weeks of gestation, can be used to detect cervical microstructure differences between patients who deliver at term and those who have inflammatory conditions and deliver preterm.

Our DBSI findings are consistent with several previous observations regarding cervix microstructures. For example, we observed the highest cell density proximal to the subglandular region. This was consistent with the known distribution of mucus-secreting cells in endocervical glands, which lie between the subglandular region and the endocervical canal[53–55]. Moreover, both ex vivo and in vivo DBSI-derived collagen fiber maps revealed a more homogeneous distribution of collagen in the subglandular region, adjacent to the cervical canal, than in the outer stroma region, as was observed previously[34,56]. Finally, DBSI identified muscle fibers that corresponded to those in trichrome images. These images revealed longitudinal fibers parallel to the cervical canal and circumferential fibers near the cervix-uterus junction, consistent with previous findings[12]. Combined with our ex vivo validation, these findings strongly indicate that our DBSI-derived measures accurately image cervix microstructures.

Our retrospective in vivo DBSI data indicated that the cervices from patients who later delivered preterm differed from those of patients who later delivered at term in three ways. First, the cervices from the preterm group had less collagen fraction than those from the term group, perhaps indicating premature collagen degradation. Additionally, the collagen fraction correlated with both days to delivery and cervical length measured by T2W MR images at 32 weeks of gestation. This finding is consistent with the observation that collagen concentration in the mouse cervix decreases over gestation[53].

Second, cervices in the preterm group exhibited greater cell fraction than those in the term group. Given that the preterm patients had all been diagnosed with infectious diseases or inflammatory conditions, the elevated cell fraction may reflect immune cell infiltration and elevated inflammation. The inverse correlation between DBSI cell

**Table 1 | Demographic characteristics of the healthy term and adverse preterm group in the in vivo studies**

| In vivo patients | | | |
|---|---|---|---|
| Demographic parameters | Healthy term | Preterm | P value |
| Number of patients | 10 | 7 | |
| Median age, years (IQR) | 31 (25–34) | 29 (27–32) | 0.82 |
| Gestational age at deilvery, weeks.days(IQR) | 39.3 (39.1–39.5) | 35.2 (34.6–36.2) | |
| Race, n | | | 0.43 |
| White | 6 | 3 | |
| African American | 3 | 4 | |
| Asian | 1 | 0 | |
| Other | 0 | 0 | |

Statistical analysis of maternal age utilized a two-sided two-sample t-test at a 0.05 significance level. Statistical test for race was analyzed using Pearson's Chi-squared test with two degrees of freedom.

fraction and both days to delivery and cervical length at 32 weeks of gestation suggest a link between elevated immune cell infiltration, a shorter and more dilated cervix, and proximity to delivery.

Third, cervices of the preterm group had higher muscle fraction than those in the term group. Additionally, the muscle fraction inversely correlated with both days to delivery and cervical length at 32 weeks of gestation. These findings suggest that patients with high muscle fraction have a shorter cervix and are closer to delivery than those with a low muscle fraction. These findings align with an observation of increased smooth muscle in cervical insufficiency[57], the idea that cervical smooth muscle contraction plays a role in dilation[58], and the hypothesis that longitudinal muscle fibers can contract and promote cervical dilation[59]. In a recent in vitro study, an ultrasound and a force transducer were used to measure contractility in cervices from non-pregnant pre-menopausal women. The effective scatterer diameter measured by the ultrasound correlated with the generated contraction force well and both measures increased immediately after administration of oxytocin[60]. Irrespective of the precise mechanisms governing cervical smooth muscle contraction and cervical dilation, the observed increase in muscle fiber density, as revealed by our study, potentially offers an anatomical substrate that could facilitate such contractions.

Together, the three DBSI-derived measures distinguished preterm and term groups, as evidenced by the absence of overlap in the two Gaussian ellipsoids. The diversity and patient-specific nature of pathways to premature cervical dilation and preterm labor suggest that the combination of these three DBSI measures might serve as a more robust predictor of preterm labor than a single measure. This hypothesis invites further investigation in future larger studies.

One DBSI measure that showed only a small difference between the term and preterm groups was the free water fraction. Additionally, the free water fraction did not correlate with either days to delivery or cervical length at 32 weeks of gestation. It is important to note that DBSI was performed at a single time point at 32 weeks' gestation, likely before a sharp increase in hydration occurs. Moreover, Maradny et al. found small differences (about 5%) in pregnant rabbit cervical hydration between control animals and those treated with hyaluronic acid to simulate cervix ripening[61]. Future longitudinal DBSI studies conducted throughout pregnancy, with a specific focus on imaging closer to delivery, might reveal a time-dependent increase in the free water fraction.

Upon establishing the DBSI method on imaging the complex microstructure throughout the entire cervix and assessing its feasibility in discerning between normal term and preterm patients, we note several limitations of this work that point to the need for future investigation. First, in the in vivo analysis, all of the patients in the

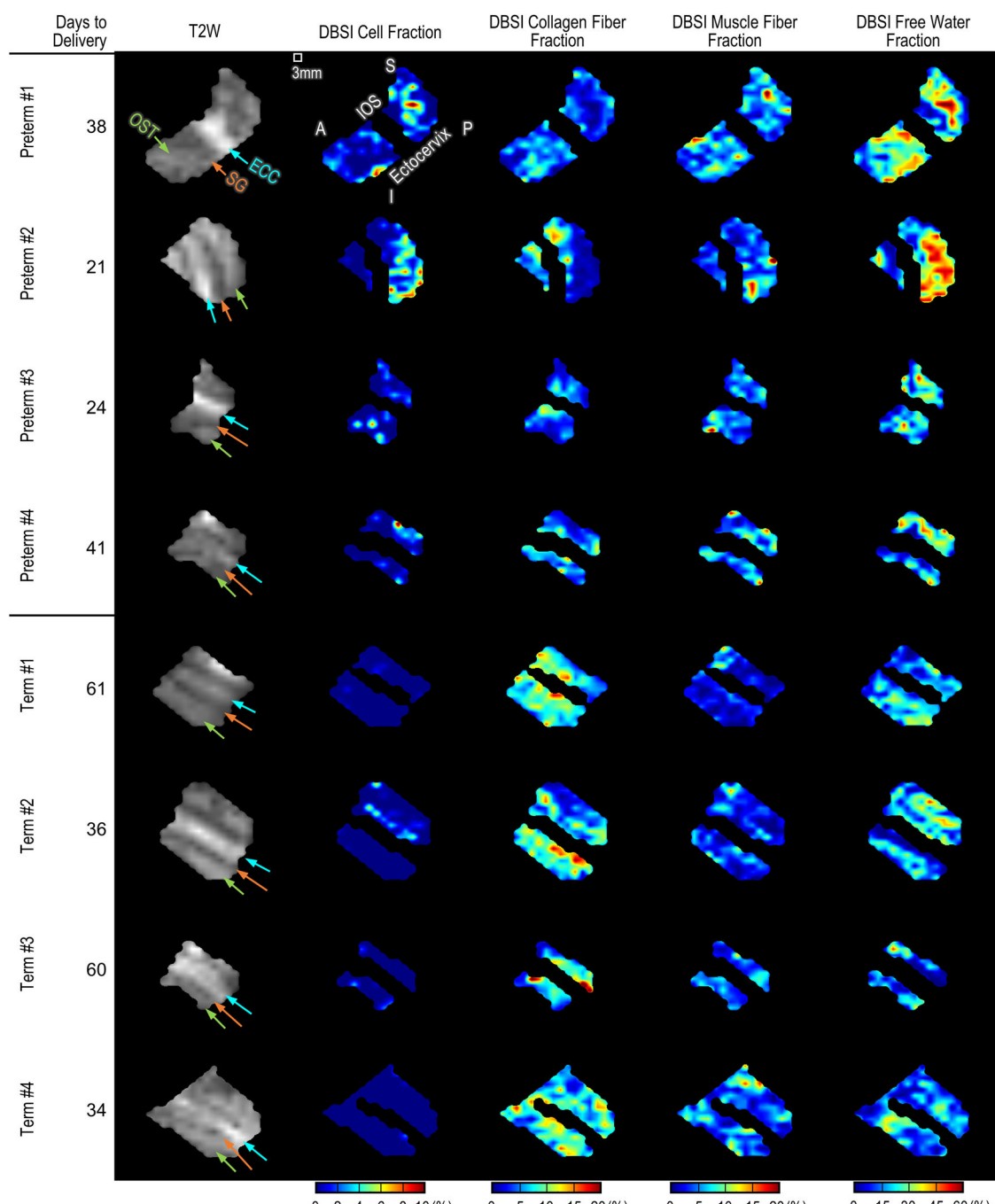

**Fig. 5 | Sagittal view of T2W images and DBSI-derived cell, collagen, muscle fraction, and free water fraction maps from representative patients in the term and preterm groups.** Days to delivery are calculated from the day of MR imaging to the date of delivery. The color-mapping is the same for all eight patients. Color-coded arrows and labels "IOS" (white), "OST" (lime green), "SG" (orange), and "ECC" (cyan blue) indicate internal OS of cervix, outer stroma, subglandular, and endo-cervical canal, respectively. Labels "S", "I", "A" and "P" indicate patient's superior, inferior, anterior, and posterior positions, respectively. Source data are provided as a Source Data file.

preterm cohort had infectious diseases or inflammatory conditions. Future studies can determine the ability of our DBSI measures to predict preterm birth in patients without any known pregnancy complications. Second, we only performed DBSI at a single timepoint. Longitudinal studies would enable us to use our DBSI measures to track patient-specific cervical remodeling. Third, in our H&E staining, we quantified nuclei density and compared it to DBSI cell fraction, which were defined as having a wide range of sizes. Future effort can be directed at developing DBSI parameters to quantitate specific immune cell types such as macrophages, etc. and validating them by using specific immunohistochemistry stains. In this way, we will obtain more specific information regarding the ability of DBSI to measure cervical inflammation.

## Methods

### Enrollment criteria for patients included in ex vivo and in vivo studies

All procedures were approved and performed in accordance with the principles of the Declaration of Helsinki and the ethical standards of the Washington University Institutional Review Board (IRB) through

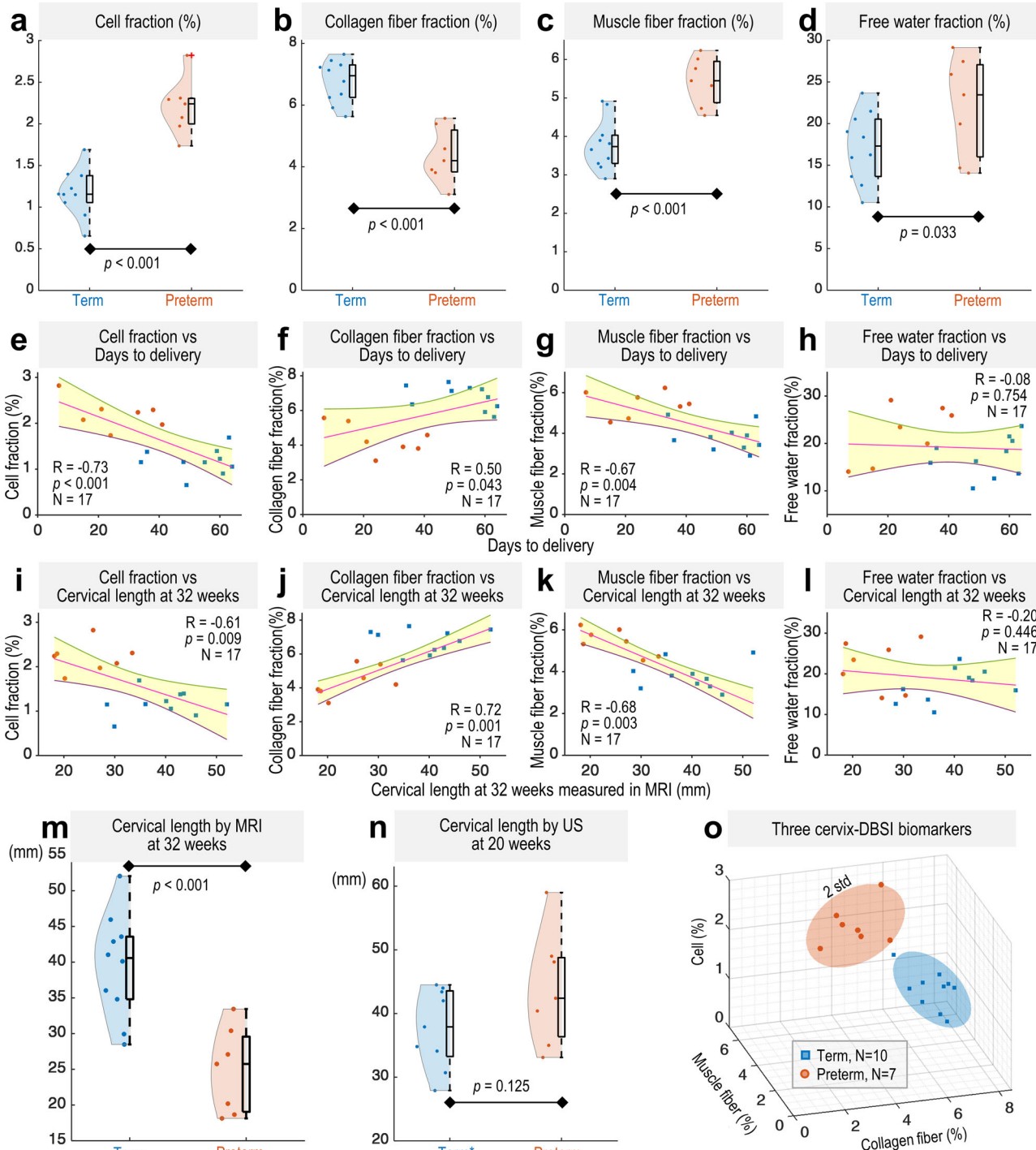

**Fig. 6 | DBSI-derived cell, collagen fiber, muscle fiber, and free water fraction in the term and preterm groups. a**–**d** Box and violin plots for DBSI measures in term (blue, $N = 10$) and preterm (orange, $N = 7$) groups, showing maxima, 75th percentile, medians, 25th percentile, and minima, alongside violin plots with each dot representing the median of nonzero DBSI values per patient across the whole cervical volume. $p = 2.87 \times 10^{-06}$, $7.35 \times 10^{-06}$, $6.30 \times 10^{-05}$ for (**a**)–(**c**), respectively. **e**–**h** Show correlations between the number of days from DBSI imaging to delivery and DBSI measures. $p = 8.01 \times 10^{-04}$ for (**e**). **i**–**l** Correlations between cervical lengths measured at $32 \pm 2$ weeks gestation in T2W MR images and DBSI measures. In (**e**)–(**l**), each blue and orange dot represents the median of nonzero DBSI values across the whole cervical volume for term and preterm patients, respectively, with red lines

for linear fits and shaded areas indicating 95% confidence intervals. **m**, **n** Cervical length at 32 weeks by MRI (**m**) and at 20 weeks by transvaginal ultrasound (**n**) for term (blue dots, $N = 10$ for (**m**); $N = 9^{*}$ for (**n**)) and preterm (orange dots, $N = 7$) groups, with box plots depicting the maxima, 75th percentile, medians, 25th percentile, and minima for each group. $p = 2.67 \times 10^{-04}$ for (**m**). *In (**n**), one term patient did not undergo transvaginal ultrasound. **o** DBSI measures visualized on X-Y-Z axes with Gaussian ellipsoids representing two standard deviations from the mean (95% probability) for each group. Statistical analysis performed with one-sided two-sample t-test at a 0.05 level, except for a two-sided test in (**n**). Correlation analysis performed with two-sided Pearson's correlation at 0.05 significance level. Source data are provided as a Source Data file.

the Human Research Protections Office. Written informed consent was obtained from all participants for the use of ex vivo specimens and derived data and in vivo data in this study (ex vivo study, HRPO: 201903056, 202006074; the parent study of in vivo data, HRPO: 201707152, 202006021). The parent study from which the in vivo data were derived was not a clinical trial, and participants received 50 USD in prepaid gift cards for each MRI visit.

Eligible pregnant participants in the ex vivo study included those carrying a single fetus with normal fetal anatomy, diagnosed with placenta accreta, planning to deliver at Barnes-Jewish Hospital (Saint Louis, Missouri), English-speaking, and aged 18 years or older. Pregnant participants were excluded if they had MRI contraindications or if the fetus presented significant anomalies. Non-pregnant participants eligible for the ex vivo study were English-speaking, aged 18 to 45 years, and scheduled for a medically-indicated or prophylactic hysterectomy at Barnes-Jewish Hospital (Saint Louis, Missouri). Exclusion criteria for non-pregnant participants included MRI contraindications, known cervical anomalies, or a diagnosis of gynecological and/or metastatic cancer.

For the parent study of in vivo data, patients were eligible if they were pregnant with a single fetus with a normal fetal anatomy and intended to deliver at Barnes-Jewish Hospital (Saint Louis, Missouri), were English-speaking, and were aged 18 years or older. Patients were excluded from the parent study if they tested positive for a blood-borne infectious disease, were an intravenous drug user, had contraindications to undergoing an MRI, had a body mass index greater than 40, or if the fetus had significant anomalies.

In this retrospective in vivo study, the following inclusion criteria were applied: For the preterm cohort with inflammation-associated adverse conditions, participants must have delivered before 37 weeks of gestation. Additionally, they must have had one or more of the following: a positive test result for gonorrhea, chlamydia, trichomoniasis, syphilis, human papillomavirus, bacterial vaginosis, yeast infection, herpes simplex virus, or COVID-19; a positive test for Group B *Streptococcus* (GBS) during pregnancy or receiving treatment for GBS; or diagnosed with endometriosis, placenta previa/accreta, or chorioamnionitis. For the healthy term cohort, participants must have delivered at or beyond 37 weeks of gestation and must not have had any of the inflammation-related adverse conditions described for the preterm group.

## Monte-Carlo simulation of water diffusion in cells, collagen, and muscle fibers

The boundary condition at the cell membrane for water molecules is specular reflection. The membrane structures inside the cell (e.g., nuclear membranes) were neglected in the modelling. The sphere models were scaled to a series of radii from 5 μm to 12 μm to mimic the size of all cells (including resident and immune cells) in the cervix. Each model had a fixed density (10 counts / μm$^3$) of random seeds (representing water molecules). A bundle of solid crossing rods, angled between 0 and 20 degrees, were used to model collagen fibers and water molecules diffusing between them to reflect various fiber orientations and dispersion patterns, characteristic of both normal and pathological conditions in the cervix. A bundle of parallel hollow tubes were used to model muscle fibers and water molecules diffusing inside them.

The simulated diffusion MRI signal was based on the same diffusion gradient (b values and b vectors) used in in vivo and ex vivo MRI. In-house Monte-Carlo simulation software (MATLAB 2022b) used the following equations:

$$l = \sqrt{6*D*dt} \qquad (1)$$

$$d\phi = \gamma G(t)*r(t)*dt \qquad (2)$$

$$S = \sum_{j=1}^{N} \mathrm{Re}(\exp(i\phi_j)) \qquad (3)$$

Where $l$ is the discretized stepsize for water molecule diffusion. $D$ is diffusion coefficient with any restriction at a certain temperature. $dt$ is the discretized time step in the simulation and is equal to 1 ms. $\gamma$ is the gyromagnetic ratio. $r(t)$ represents the full trajectory of water molecule random motion within the simulation duration. $G(t)$ is the bipolar diffusion gradient. $\phi_j$ is the dephase of the $j^{th}$ water molecule. $S$ is the normalized simulated diffusion signal.

## Cervix-optimized DBSI multi-tensor model

DBSI[44] uses Eq. 4 to model diffusion-weighted MRI signals in each voxel as a linear combination of multiple anisotropic and isotropic tensors that are tailored to microstructures of the specific organ and disease model.

$$S_k = \underset{\text{Total signals}}{\sum_{i=1}^{N_{Aniso}} f_i e^{-\left|\vec{\mathbf{b}}_k\right| \cdot \lambda_{\perp i}} e^{-\left|\vec{\mathbf{b}}_k\right| \cdot (\lambda_{\|i} - \lambda_{\perp i}) \cdot \cos^2 \psi_{ik}}}$$
$$\text{Signals from anisotropic tensors}$$
$$+ \underset{\text{Signals from isotropic tensors}}{\sum_{j=1}^{N_{Iso}} f_j e^{-\left|\vec{\mathbf{b}}_k\right| D_j}} (k=1,2,\ldots,K) \qquad (4)$$

$S_k$ is the diffusion-weighted signal at each voxel, and $\left|\vec{\mathbf{b}}_k\right|$ is the b value of the $k^{th}$ diffusion gradient. $N_{Aniso}$ and $N_{Iso}$ represent the number of anisotropic and isotropic tensors, respectively. $\psi_{ik}$ denotes the angle between the $k^{th}$ diffusion gradient and the principal direction of the $i^{th}$ anisotropic tensor. $\lambda_{\|i}$ and $\lambda_{\perp i}$ are the axial diffusivity and radial diffusivity, respectively, of the $i^{th}$ anisotropic tensor. $f_i$ is the signal intensity fraction for the $i^{th}$ anisotropic tensor, and $f_j$ is the signal intensity fraction for the $j^{th}$ isotropic tensor. $D_j$ denotes isotropic diffusivity of $j^{th}$ isotropic tensor.

The principal directions $\psi_{ik}$ of the anisotropic tensors were designed based on a set of uniformly distributed unit vectors in a 3D space[46]. The AD and RD of anisotropic tensors were designed based on the Monte-Carlo simulation results. The nine detailed anisotropic tensor models of AD and RD combinations are described in the supplementary materials (Table S1). In Table S1, the tensor models #2 and #3 (AD = 1.8 – 2.0 × 10$^{-3}$ mm$^2$/s RD = 0.3 – 0.4 × 10$^{-3}$ mm$^2$/s at 37 °C) represent muscle fibers, and the tensor models #5, #6, and #7 (AD = 1.6 – 2.1 × 10$^{-3}$ mm$^2$/s RD = 0.5 – 0.7 × 10$^{-3}$ mm$^2$/s at 37 °C) represent collagen fibers. The other tensor models represent remaining connective tissue structures in the cervix. All these tensor models were replicated in 25 principal directions uniformly distributed in a 3D space to achieve a total of 225 anisotropic tensors.

Three types of isotropic tensors (Fig. 1) were designed to describe cell water, hindered water, and free water in an image voxel. Restricted isotropic diffusion tensors (diffusivity = 0 – 0.6 × 10$^{-3}$ mm$^2$/s at 37 °C, Fig. S13) were designed for water molecules inside cell membranes with diameters less than 12 μm in the Monte-Carlo simulation. Diffusivity between 3 – 10 × 10$^{-3}$ mm$^2$/s at 37 °C was chosen for free water diffusion and is based on experimental values by Easteal et al. and Holz et al.[51,52]. Hindered isotropic diffusion tensors (diffusivity = 0.6 – 3 × 10$^{-3}$ mm$^2$/s at 37 °C) were designed for water molecules hindered by the dense, organized collagen fiber and cells. Additionally, diffusivity exceeding 10 × 10$^{-3}$ mm$^2$/s was utilized to account for intravoxel incoherent motion effect. A total of 150 isotropic tensors (Fig. S13) were then designed by discretizing all four ranges of isotropic diffusivity values. For ex vivo specimens imaged at 20 °C, a linear coefficient of 0.6667 was applied on all isotropic and anisotropic diffusivity values according to quadratic fit of known diffusivity values at a range of temperatures[51,52].

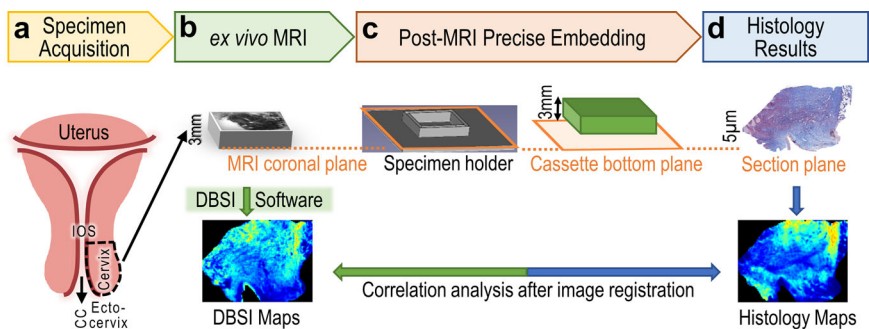

**Fig. 7 | Experimental set-up to use ex vivo specimens to validate cervix-optimized DBSI parameters. a** A specimen (3 mm thickness) was dissected from the posterior midline of fresh total hysterectomy uteruses. Labels "IOS" and "ECC" indicate internal OS of cervix and endocervical canal respectively. **b** The specimen was embedded in 2% agar gel and imaged with a Varian 11.7 T MRI, using DBSI sequence. DBSI maps were computed by in-house cervix-optimized DBSI software. **c** After MRI, the specimen was fixed in formalin, transferred, and embedded in a paraffin block along the plane of MR imaging (orange dotted lines indicates the same plane). **d** The stained slide-mounted 5 μm histologic sections were digitized and converted to histology maps. The histology maps were then registered to the DBSI maps.

## Ex vivo imaging of cervical tissue blocks

Blocks of tissue (approximately 30 × 25 × 3 mm³) were dissected from the posterior midline of fresh total hysterectomy specimens. A custom specimen holder (Fig. S14) was 3D printed (Form Labs, Form 2, RS-F2-GPCL-04 clear resin). The tissue block was embedded in 2% agar gel (Fig. 7b) (Sigma, SKU-0504D-100G) in the specimen holder at approximately 40 °C and quickly cooled to room temperature. It was then placed in the temperature-controlled MRI scanner room at 20 °C for 30 min to allow the specimen temperature to reach the ambient temperature. The specimen holder was inserted into a custom-made Helmholtz pair coil (Extend MR, LLC, Fig. S15). MR images were acquired in an Agilent/Varian 11.74-T (500-MHz), DirectDrive MRI scanner (United States) with Vnmrj console software version 4.2. Several scout images were acquired, and the specimen holder was adjusted to align the MRI bottom coronal plane with the specimen holder bottom plane. The specimen was then imaged by using two-dimensional single-shot spin-echo sequences for both T2W images and diffusion image series with the following parameters: Repetition time (TR) = 1000 ms, echo time (TE) = 32 ms, FOV = 36 × 36 mm, data matrix = 144 × 144, slice thickness = 1.0 mm, no slice spacing. For the diffusion images, the diffusion gradients were applied in up to 74 directions (same as in vivo, see Table S2) with $b$ values of 0 – 4500 s/mm². After imaging, the specimens were fixed in 10% neutral-buffered formalin for one week and then processed into paraffin in "as-is" orientation. Sections (5 μm) were cut from the bottom plane, which was the same as the MRI bottom coronal plane, mounted on slides, and stained with hematoxylin & eosin and Masson's trichrome (Ventana Trichrome Kit (92)860-031). The slide images were digitized (20× magnification, Hamamatsu NanoZoomer HT).

## Histology quantification

A 20 × bright field image was trimmed to tiles (5000 × 5000 pixels) and stitched together after processing. Researchers with histology training first delineated about a dozen typical cell nuclei, regions of background, and connective tissue (eosin stain) on one tile image to train an Orbit Image Analysis[62] tissue classifier based on chromatic information. All image tiles were classified to binary masks of positive nuclei stain by the trained classifier. Then, the remainder of the histology quantification was done by an in-house program computed in MATLAB 2022b. The binary masks were converted to nuclei density maps by calculating percentage of nuclei area per 50 × 50 pixels area. These tiles of nuclei density maps were then stitched together and downsampled to 10 times the MR image plane resolution for the next step of image registration. The collagen fiber and muscle fiber density maps were processed with the same pipeline by classifying chromatic information of positive collagen stain (blue) and muscle stain (scarlet) in trichrome histology.

## Image registration

The specimen contours from the nuclei, collagen fiber, and muscle fiber density maps and T2W MR image were delineated in 3D slicer version 5.2.1[63]. An Affine linear registration guided by fiducial landmarks was performed to register the nuclei density map, collagen fiber fraction map, and muscle fiber fraction map to the T2W MR image. After registration, the histology maps were down sampled to the same resolution as the DBSI-derived maps. In the correlation analysis, both DBSI and histology maps were further downsampled to 2.5 mm to match the resolution of in vivo diffusion MR images from a clinical MRI scanner.

## Human in vivo MR imaging

MR images from were acquired from a Siemens 3 Tesla MAGNETOM Vida (Erlangen, Germany) at patients' gestational ages of 32 ± 2 weeks. Patients were imaged in the left lateral position with their feet entering the magnet bore first. A 30-channel phased array torso coil covered the entire pelvis. Each patient underwent the following MRI sequences: 1) a set of localizer images to adjust the field of view to cover the entire uterus and cervix; 2) T1-weighted volume-interpolated breath hold examination sequence in the sagittal plane (see Supplements for detailed parameters); 3) T2-weighted turbo spin echo high-resolution images in sagittal planes; 4) T2-weighted turbo spin echo high-resolution images in oblique planes perpendicular to the cervical canal (see Supplements for detailed parameters); and 5) two-dimensional single-shot echo planar imaging diffusion-weighted sequence, with the following parameters: repetition time (TR), 14600 ms; echo time (TE), 62 ms; field of view (FOV), 384 × 384 mm; data matrix, 128 × 128; slice thickness, 3.0 mm; no slice spacing. The diffusion gradients were applied in up to 74 directions (same as ex vivo, see Table S2 for $b$ tables) with $b$ values ranging of 0–2000 s/mm². The total imaging time was up to 45 min.

## Statistics and reproducibility

In the ex vivo study, diffusion MR images were acquired once to preserve the tissue integrity of fresh specimens, with each image voxel representing an independent measurement. Bright-field images of 5 μm histologic sections (Figs. 2a–4a) were captured with a Hamamatsu NanoZoomer following manufacturer's calibration instructions. For spatial correlation analyses, both DBSI and histological maps were aligned to 2.5 × 2.5 mm grid boxes to match the resolution of clinical in vivo diffusion MR images. The number of grid boxes was specimen

size-dependent. Additionally, correlation analyses across various grid sizes (Figs. S1–S4 in supplementary materials) consistently demonstrated a strong correlation between histology quantification and ex vivo DBSI measures, affirming this method's reliability. Spatial correlations were assessed using two-sided Pearson's correlation at a 0.05 significance level.

In the in vivo study, an initial cohort of ten preterm patients was reduced to seven after excluding three for inadequate image quality and severe motion blur. This study aims to develop, validate, and demonstrate the clinical feasibility of an imaging method, negating the need for randomization and blinding. Maternal age and race comparisons between term and preterm groups used a two-sided two-sample t-test at a 0.05 significance level and Pearson's Chi-squared test with two degrees of freedom, respectively. Differences between term and preterm groups in DBSI measures and cervical length at 32 weeks were evaluated using one-sided two-sample t-tests at a 0.05 level, except for cervical length via transvaginal ultrasound at 20 weeks, which used a two-sided test. Correlations between DBSI measures with days to delivery and cervical length at 32 weeks were determined using two-sided Pearson's correlation at a 0.05 significance level.

### Reporting summary
Further information on research design is available in the Nature Portfolio Reporting Summary linked to this article.

## Data availability
The data supporting the findings of this study are available in the article and its Supplementary information. DBSI and histology maps have been deposited at Figshare: https://doi.org/10.6084/m9.figshare.25584081[64]. Source data are provided with this paper.

## Code availability
The DBSI code that supports the findings of this study is available upon request at Zenodo https://doi.org/10.5281/zenodo.11085858[65]. Code's functionality is described on the deposit site.

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

## Acknowledgements

This study was supported, in part, by the March of Dimes Center Grant (22-FY14-486), by grants from National Institutes of Health (NIH) / National Institute of Child Health and Human Development (R01HD094381 to PIs Y. Wang/Cahill; R01HD104822 to PIs Y.Wang / Schwartz / Cahill), by grants from the National Institutes on Aging (R03AG072375-01 and R01AG074909 to PI Q. Wang), by grants from Burroughs Wellcome Fund Preterm Birth Initiative (NGP10119 to PI Y. Wang) and by grants from Bill & Melinda Gates Foundation (INV-005417, INV-035476, and INV-037302 to PI Y. Wang). We thank the Alafi Neuroimaging Laboratory for allowing us to use their Hamamatsu Nano-Zoomer (funded by NIH shared instrumentation grant S10-RR0227552) to digitize histology images. We thank Jessica Chubiz for managing the study participants. We thank Stephanie Pizzella, Jordyn Lehr, Lara

Goodrich, Bri Dawson, Eva Goins, Madison Copeland, Jessica Battle, Claire Novack, Cassy Hardy, Emily Crews, Monica Anderson, Tracy Burger, Emily Diveley and Megan Steiner for explaining the study to potential participants, obtaining consent, and managing the study. We thank Scott Love, BSRT, Mark Nolte, RT, and Glenn Foster, RT, for conducting clinical MR imaging; Jim Quirk, PhD, for guidance on operating the Varian 11.7 T small animal MRI; Deborah Frank, PhD, and Ian S. Hagemann, MD, PhD, for critical review of the manuscript; Sue Boss-Miller and Jessie Archie for collecting specimens; Zulfia Kisrieva-Ware for wet lab preparation; Alma Johnson for histology sectioning; Phillip Foxwell and team for trichrome staining; Autumn Watson and AMP team for H&E staining; and Wenxu Qi, MD, PhD for literature review.

## Author contributions

W.W., Z.S., Q.W., and Y.W. designed the experiments and the methods. W.W., Z.S., H.G., Y.N., Q.W., and Y.W. contributed to the development of the cervix DBSI processing pipeline. W.W. and Z.S. developed and optimized the ex vivo MRI sequences and scans. W.W., Z.S., and H.X. conducted ex vivo experiments. Q.W. developed and optimized the human in vivo MRI sequences and scans. W.W., Z.S., H.G., Y.N., H.X., J. L., Y.L., H.W., conducted human in vivo experiments. S.P., P.K.W., H.R.K., contributed to the study design and guided the clinical studies. W.W. and Z.S. processed and analyzed the data.

## Competing interests

The authors declare no competing interests.
