## [Peer Review File · Nature Communications]

REVIEWER COMMENTS

Reviewer #1 (Remarks to the Author):

Whole cervix imaging of collagen, muscle, and cellularity in term and preterm birth

Review:

In this paper, the authors propose to use DBSI, a diffusion MRI modeling technique, to correlate and validate dBSI derived measures for predicting preterm birth due to early dilation of the cervix.

The authors have done an excellent job at describing the background material and where the field stands currently. The issue of early prediction of pre-term birth can be quite useful as described in this paper.

While the authors have done an excellent job of doing histology based validation of their work along with some simulation experiments, several aspects are quite unclear as listed below:

1. In Figs. 2, 3 and 4 — a user defined box or region is chose to show high correlation between histology and DBSI measures. However, this doesn't really show the true picture as it artificially boosts the correlation. Ideally, if DBSI is highly sensitive to the cell/fiber density, then the whole image should be correlated. As such, I would request that the authors show correlation for the entire image rather than only a few hand-picked boxes.
2. It is not clear if a similar (user chosen box ROI) approach was used in Fig 6 to find group differences and correlations? Were the entire images used or only a particular hand-picked slice? Please clarify what was done and what criteria were used to choose certain regions of interest. The only part that is clear is that a mask was used for the mucus-filled endocervica canals. No other aspects are clear (choosing specific slices or regions for correlation analysis).
3. Finally, while DBSI model was updated to use several isotropic and anisotropic compartments, the ranges of the diffusivities were fixed. These ranges were derived from a normative model of the cervix tissue. However, since the paper wants to estimate "abnormalities" in the diffusivities and fractions of the cervix in preterm, how do fixing these ranges in DBSI affect the estimates of the fractions?

Essentially — my question is — what if the diffusivities in the preterm cervix tissue does not fall in the proper range of values as defined my simulation experiments? Can some tests be done to show how the estimation works/fails in this case?

4. It will also be useful to see some comparison with at-least a couple of models: e.g. DTI and diffusion propagator model or MAP-MRI model? This will ensure that these results are specific to DBSI.

5. The authors have provided in the supplementary material the b-values and the gradient directions used. It seems that quite a few gradients at very low b-values were used. This raises the issue of blood flow (i.e. IVIM effect) being a significant contributor to the signal at such low b-values that was not accounted for in the model. Perhaps that itself could be marker for cervix dilation which should be investigated.

Once all of these comments are addressed, I would be comfortable to recommend this paper for possible publication.

Reviewer #2 (Remarks to the Author):

This study has two parts. First, ex vivo human cervical tissue samples are measured for cellularity, collagen fibers, and muscle fibers using histologic and MRI methods. Second, a retrospective study was conducted to investigate whether cervical cellularity, collagen fibers, and muscle fibers are related to spontaneous preterm birth in human pregnancy using a previously collected MRI dataset of pregnant patients. To accomplish this study, a magnetic resonance diffusion basis spectrum imaging (DBSI) technique was developed to measure these cervical quantities. There is high confidence in the methodology, as it has been developed for the whole brain and validated for specific cases of brain inflammation in multiple sclerosis in another study. The presented manuscript validated the DBSI method on whole ex vivo human cervical tissue samples excised from hysterectomy specimens. DBSI signals for collagen, cells, and muscle fibers were confirmed with histology. Then, this DBSI method was applied to previously acquired MRIs from a large-scale March of Dimes-funded clinical study to understand if there are distinct differences in cellularity, collagen fibers, and muscle fibers between patients who delivered at term compared to patients who delivered before 37 weeks. This paper is well-written, with precise figures of cervical microstructural signals and explanations of the DBSI method. The data presented here would be of broad interest to the readership of this journal as it presents critical mechanistic data to understand cervical remodeling in pregnancy. Particularly, highlighting how advanced medical physics methods can be applied to topics in maternal health would also be of high

interest to the readership. The paper attempts to make a concise summary of cervical remodeling in the introduction. There are a few corrections the authors should consider before publication. See my comments below:

Comments:

1. Please consider changing the first few statements of the introduction, implying the cervix remains stiff throughout the majority of pregnancy and then softens near the end. Two lines of evidence in human studies suggest the cervix begins to soften sooner than that – see [1. Badir S, Mazza E, Zimmermann R, Bajka M. Cervical softening occurs early in pregnancy: characterization of cervical stiffness in 100 healthy women using the aspiration technique. *Prenat Diagn.* 2013 Aug;33(8):737–741. PMID: 23553612 and 2. Carlson LC, Hall TJ, Rosado-Mendez IM, Mao L, Feltovich H. Quantitative assessment of cervical softening during pregnancy with shear wave elasticity imaging: an in vivo longitudinal study. *Interface Focus.* 2019 Oct 6;9(5):20190030. PMID: PMC6710662]. Additionally, rodent data also suggest the cervix softens in stages, a summary of rodent data is here [1. Yoshida K, Jayyosi C, Lee N, Mahendroo M, Myers KM. Mechanics of cervical remodeling: insights from rodent models of pregnancy. *Interface Focus.* Royal Society; 2019 Oct 6;9(5):20190026.].
2. In the introduction, establish for the reader what the cervix is made of – what are the extracellular matrix components, and what are the cells?
3. Are you measuring collagen content per dry or wet weight? Please make a specification.
4. Please comment on the practicality of using MRI in a clinical diagnosis setting. Additionally, measurements are taken at 32 weeks of gestation. Many patients with cervix-related spontaneous birth deliver earlier. Do you recommend a specific gestation window for this diagnostic measure? Do you have plans to study the cervix at earlier timepoints typical for cervical length screening in the clinic?

REVIEWER COMMENTS

Reviewer #1 (Remarks to the Author):

Whole cervix imaging of collagen, muscle, and cellularity in term and preterm birth

Review:

In this paper, the authors propose to use DBSI, a diffusion MRI modeling technique, to correlate and validate DBSI derived measures for predicting preterm birth due to early dilation of the cervix.

The authors have done an excellent job at describing the background material and where the field stands currently. The issue of early prediction of pre-term birth can be quite useful as described in this paper.

While the authors have done an excellent job of doing histology based validation of their work along with some simulation experiments, several aspects are quite unclear as listed below:

1. In Figs. 2, 3 and 4 — a user defined box or region is chosen to show high correlation between histology and DBSI measures. However, this doesn't really show the true picture as it artificially boosts the correlation. Ideally, if DBSI is highly sensitive to the cell/fiber density, then the whole image should be correlated. As such, I would request that the authors show correlation for the entire image rather than only a few hand-picked boxes.

Respond: We appreciate the reviewer's insightful comments, which allow us to clarify the methodology behind the correlation analysis in Figs. 2, 3, and 4. The high correlation presented in the original manuscript encompasses the entire specimen, not just the areas within the black dashed boxes. These black dashed boxes in Panels B and C are mere locational markers for the corresponding 10x histology images shown in Panel A, and do not singularly constitute our correlation analysis basis.

To enhance clarity, revised Figs. 2–4 now feature white grids overlaid on the maps in Panels B and C. Each blue dot in Panel D represents the mean value derived from a 2.5 x 2.5 mm white grid box, correlating histology quantification with *ex vivo* DBSI measures. Data from all grid boxes covering the entire tissue are employed in the correlation analysis shown in Panel D of Figs. 2–4.

Furthermore, the chosen 2.5 x 2.5 mm grid size aligns with standard *in vivo* diffusion MRI resolutions in clinical MR scanners. To reinforce the validity of our findings, we performed additional analyses with varying grid sizes (2 x 2 mm, 1.75 x 1.75 mm, and 1.5 x 1.5 mm), as documented in the revised supplementary materials (Fig S. 1–Fig S. 4). These analyses consistently demonstrate a strong correlation between histology quantification and *ex vivo* DBSI measures, underscoring the robustness of our approach.

It is also pertinent to acknowledge the technical challenges inherent in aligning quantitative histology images (Panel B) with *ex vivo* DBSI images (Panel C). These include unavoidable tissue deformation during histological processing (non-uniform shrinkage during fixation, embedding, and sectioning) and discrepancies in slice thickness between histology (5 μ m) and MRI (1000 μ m) slices. These factors limit the feasibility of further reducing grid size in our analysis, while not detracting from the validity of our imaging findings.

Fig S.2: Pearson correlation analysis of DBSI versus histology at grid sizes of 2.5mm, 2mm, 1.75mm, and 1.5mm for specimen P1-S2. Data points indicate mean values per grid box from Fig. 2, 3, and 4, panels B and C. Red lines: linear fits; shaded areas: 95% confidence intervals.

2. It is not clear if a similar (user chosen box ROI) approach was used in Fig 6 to find group differences and correlations? Were the entire images used or only a particular hand-picked slice? Please clarify what was done and what criteria were used to choose certain regions of interest. The only part that is clear is that a mask was used for the mucus-filled endocervical canals. No other aspects are clear (choosing specific slices or regions for correlation analysis).

Respond: We regret any misunderstanding caused by our initial presentation. The *in vivo* data depicted in Fig. 6 were obtained from comprehensive DBSI analyses of the entire cervix volume, with the exclusion of the cervical canal, across all imaging slices. Each data point in Fig. 6 represents the median value of DBSI-derived measure across the entire cervical volume of a patient, excluding the cervical canal.

To address this confusion and provide greater clarity, we have amended the manuscript accordingly (Page 11, Line 34; Page 14, Line 3).

3. Finally, while DBSI model was updated to use several isotropic and anisotropic compartments, the ranges of the diffusivities were fixed. These ranges were derived from a normative model of the cervix tissue. However, since the paper wants to estimate “abnormalities” in the diffusivities and fractions of the cervix in preterm, how do fixing these ranges in DBSI affect the estimates of the fractions?

Essentially — my question is — what if the diffusivities in the preterm cervix tissue does not fall in the proper range of values as defined by my simulation experiments? Can some tests be done to show how the estimation works/fails in this case?

Respond: We are grateful for the reviewer's insights. The DBSI methodology effectively models a broad spectrum of anisotropic and isotropic compartments (375 tensors in total) to accurately represent both physiological and pathological microstructures within the cervix. Specifically, Our DBSI tensor models are intricately designed based on Monte Carlo simulations that reflect the realistic microstructural features of the cervix, encompassing a variety of cellular structures (including resident and immune cells) as well as extracellular matrix components such as collagen and muscle fiber bundles in different amounts and architectures. These models are specifically designed to reflect various fiber orientations and dispersion patterns indicative of both normal and pathological conditions (Page 4, Line 40; Page 5, Line 14; Page 17, Line 7).

As illustrated in Fig. 6, the distribution of DBSI-derived measures across different DBSI compartments mirrors the underlying pathophysiology in two patient groups. Our models are meticulously designed to capture the range of microstructural variations characterizing both normal and pathological states in cervical tissue. Notably, the cervical specimens analyzed in this study (P1-S1, P1-S2, P2-S1) were obtained from pregnant patients with preterm birth complications. The quantitative histological validation presented in Figs. 2 – 4 supports DBSI's capability to quantify cervical pathologies in these samples.

We have made appropriate revisions in the manuscript (Page 4, Line 40; Page 5, Line 14; Page 11, Line 19; Page 17, Line 9; Page 18, Line 15) to enhance clarity on these aspects.

4. It will also be useful to see some comparison with at-least a couple of models: e.g. DTI and diffusion propagator model or MAP-MRI model? This will ensure that these results are specific to DBSI.

Respond: We acknowledge and appreciate the reviewer's constructive suggestion. Pursuant to this advice, we have incorporated and applied the established methodologies of Diffusion Tensor Imaging (DTI) and Free Water Imaging (FWI), as outlined by Bergamino (2021), to our dataset. A representative example is provided in Fig S. 6 and Fig S. 10 below, with full results detailed in Fig S. 5 – Fig S. 12 of the Supplementary Materials.

Our analysis utilizing the DTI model (Fig S. 5 – Fig S. 8, all specimens) reveals that the histology-derived data, including nuclei density indicative of cellularity and collagen fiber density, do not consistently correlate with DTI measures such as apparent diffusion coefficient (ADC), fractional anisotropy (FA), and axial (AD) and radial diffusivity (RD). Notably, muscle density derived from histology shows a partial correlation with FA. However, given that DTI FA represents the averaged diffusion anisotropy from the entire imaging voxel, encompassing both collagen and muscle fibers, it lacks specificity as a biomarker for cervical collagen and muscle fibers. Similarly, in the Free Water Imaging analysis (Fig S. 9 – Fig S. 12, all specimens), we observe similar findings. The histological data do not show a reliable correlation with FWI measures. In conclusion, the DBSI methodology effectively models the intricate heterogeneity of human cervical microstructure, enabling a highly specific quantification of diverse pathologies within the human cervix.

Ref: M. Bergamino, R. R. Walsh, A. M. Stokes, Free-water diffusion tensor imaging improves the accuracy and sensitivity of white matter analysis in Alzheimer's disease. *Sci. Reports* 2021 11:11, 1–12 (2021).

Fig S. 6: Pearson correlation analysis of DTI versus histology for specimen P1-S2. Data points indicate mean values per grid box from Fig. 2, 3, and 4, panels B and C. Magenta lines: linear fits.

Fig S. 10: Pearson correlation analysis of FWI versus histology for specimen P1-S2. Data points indicate mean values per grid box from Fig. 2, 3, and 4, panels B and C. Magenta lines: linear fits.

5. The authors have provided in the supplementary material the b-values and the gradient directions used. It seems that quite a few gradients at very low b-values were used. This raises the issue of blood flow (i.e. IVIM

effect) being a significant contributor to the signal at such low b-values that was not accounted for in the model. Perhaps that itself could be marker for cervix dilation which should be investigated.

Once all of these comments are addressed, I would be comfortable to recommend this paper for possible publication.

Respond: We concur with the reviewer's observations regarding intravoxel incoherent motion (IVIM). Indeed, the DBSI methodology incorporates perfusion effects by assigning signals with isotropic diffusivity exceeding $10 \times 10^{-3} \text{ mm}^2/\text{s}$ to flow and perfusion effects, as shown in Fig S. 13. This threshold was chosen based on the established diffusion MRI literature. We have revised the manuscript (Page 5, Line 6; Page 18, Line 23) to reflect this aspect more clearly.

We acknowledge that integrating low b-value diffusion-weighted images could facilitate an in-depth exploration of the relationship between cervical dilation and blood perfusion. We intend to address this area in future research project, aiming for a comprehensive examination between cervix blood perfusion and cervix dilation.

Fig S. 13: Isotropic tensor models for whole cervix DBSI

Reviewer #2 (Remarks to the Author):

This study has two parts. First, *ex vivo* human cervical tissue samples are measured for cellularity, collagen fibers, and muscle fibers using histologic and MRI methods. Second, a retrospective study was conducted to investigate whether cervical cellularity, collagen fibers, and muscle fibers are related to spontaneous preterm birth in human pregnancy using a previously collected MRI dataset of pregnant patients. To accomplish this study, a magnetic resonance diffusion basis spectrum imaging (DBSI) technique was developed to measure these cervical quantities. There is high confidence in the methodology, as it has been developed for the whole brain and validated for specific cases of brain inflammation in multiple sclerosis in another study. The presented manuscript validated the DBSI method on whole *ex vivo* human cervical tissue samples excised from hysterectomy specimens. DBSI signals for collagen, cells, and muscle fibers were confirmed with histology. Then, this DBSI method was applied to previously acquired MRIs from a large -scale March of Dimes-funded clinical study to understand if there are distinct differences in cellularity, collagen fibers, and muscle fibers between patients who delivered at term compared to patients who delivered before 37 weeks. This paper is well-written, with precise figures of cervical microstructural signals and explanations of the DBSI method. The data presented here would be of broad interest to the readership of this journal as it presents critical mechanistic data to understand cervical remodeling in pregnancy. Particularly, highlighting how advanced medical physics methods can be applied to topics in maternal health would also be of high interest to the readership. The paper attempts to make a concise summary of cervical remodeling in the introduction. There are a few corrections the authors should consider before publication. See my comments below:

Comments:

1. Please consider changing the first few statements of the introduction, implying the cervix remains stiff throughout the majority of pregnancy and then softens near the end. Two lines of evidence in human studies suggest the cervix begins to soften sooner than that – see [1. Badir S, Mazza E, Zimmermann R, Bajka M. Cervical softening occurs early in pregnancy: characterization of cervical stiffness in 100 healthy women using the aspiration technique. *Prenat Diagn.* 2013 Aug;33(8):737–741. PMID: 23553612 and 2. Carlson LC, Hall TJ, Rosado-Mendez IM, Mao L, Feltovich H. Quantitative assessment of cervical softening during pregnancy with shear wave elasticity imaging: an *in vivo* longitudinal study. *Interface Focus.* 2019 Oct 6;9(5):20190030. PMID: PMC6710662]. Additionally, rodent data also suggest the cervix softens in stages, a summary of rodent data is here [1. Yoshida K, Jayyosi C, Lee N, Mahendroo M, Myers KM. Mechanics of cervical remodeling: insights from rodent models of pregnancy. *Interface Focus. Royal Society;* 2019 Oct 6;9(5):20190026.]

Respond: We appreciate the advice provided by the reviewer and recognize the importance of accurately depicting cervical biomechanics throughout gestation. Accordingly, we have revised the introductory section of our manuscript (Page 3, Line 1) to reflect the evidence suggesting that cervical softening may occur earlier and process in stages in pregnancy than previously described. This update is now substantiated by the inclusion of references to the work by Badir et al. (2013), Carlson et al. (2019), and Yoshida et al. (2019), which provide compelling evidence of progressive cervical softening as early as the first trimester and throughout the majority of the third trimester.

These modifications not only enrich the context of our study but also align our manuscript with the current understanding of cervical biomechanics. We are confident that these revisions amplify the manuscript's scientific accuracy and depth.

2. In the introduction, establish for the reader what the cervix is made of – what are the extracellular matrix components, and what are the cells?

Respond: We appreciate this suggestion and now reorganized the introduction (Page 3, Line 6) to include the basic composition of the cervix (ECM and cells).

“The cervix is composed of an extracellular matrix (ECM) containing fibrillar collagen (types I and III), elastin, and proteoglycans, alongside cellular components including fibroblasts, smooth muscle cells, and immune cells. This composition is critical for the cervix's structural integrity. Progressive remodeling of the cervical ECM leads to restructured collagen fibrils and a consequent reduction in tensile strength. Predominantly, quantitative assessments of cervical remodeling have focused on variations in collagen, smooth muscle, and cellularity.”

3. Are you measuring collagen content per dry or wet weight? Please make a specification.

Respond: We are grateful for the reviewer's comments. In diffusion-weighted (DW) MRI, the raw signal encompasses a spectrum of T2-weighted signals captured under varying diffusion gradients. DBSI, as an advanced diffusion MRI technique, quantifies collagen content by calculating the fraction of the T2-weighted signal attributable to water molecules interspersed among collagen fibers, relative to the total T2-weighted signal intensity from the entire imaging voxel. Hence, the DBSI-derived collagen fraction provides an index of collagen content in relation to the T2 signal intensity, as observed in the b0 image (an image with zero diffusion weighting). We have updated our manuscript (Page 11, Line 19) to elucidate this concept for enhanced clarity.

4. Please comment on the practicality of using MRI in a clinical diagnosis setting. Additionally, measurements are taken at 32 weeks of gestation. Many patients with cervix-related spontaneous birth deliver earlier. Do you recommend a specific gestation window for this diagnostic measure? Do you have plans to study the cervix at earlier timepoints typical for cervical length screening in the clinic?

Respond: In considering the practical application of the DBSI technique for clinical diagnosis, it's noteworthy that DBSI employs diffusion MRI sequences that are already standard in most clinical MRI systems. The duration of a diffusion scan is typically 15 – 20 minutes, making it a feasible option in a clinical setting.

Regarding the timing of imaging and early spontaneous preterm births, we recognize the significance of this issue. Following the rigorous validation as demonstrated in our current study, we propose that DBSI screenings could feasibly be performed as early as the first and second trimesters, coinciding with routine cervical length screening appointments, with the possibility of follow-up scans in the third trimester up to 37 weeks, as needed.

REVIEWERS' COMMENTS

Reviewer #1 (Remarks to the Author):

The authors have now clearly addressed all my concerns regarding the correlations shown. The figures have also been clarified. I therefore recommend to accept this manuscript.

Reviewer #2 (Remarks to the Author):

The authors addressed this reviewer's concerns.